# Prognostic Accuracy of CTP Summary Maps in Patients with Large Vessel Occlusive Stroke and Poor Revascularization after Mechanical Thrombectomy—Comparison of Three Automated Perfusion Software Applications

Iris Muehlen [1,]*, Matthias Borutta [2], Gabriela Siedler [2], Tobias Engelhorn [1], Stefan Hock [1], Michael Knott [1], Philip Hoelter [1], Bastian Volbers [2], Stefan Schwab [2] and Arnd Doerfler [1]

[1] Department of Neuroradiology, University Hospital Erlangen, Friedrich-Alexander-University of Erlangen-Nuremberg (FAU), 91054 Erlangen, Germany; tobias.engelhorn@uk-erlangen.de (T.E.); stefan.hock@uk-erlangen.de (S.H.); michael.knott@uk-erlangen.de (M.K.); philip.hoelter@uk-erlangen.de (P.H.); arnd.doerfler@uk-erlangen.de (A.D.)

[2] Department of Neurology, University Hospital Erlangen, Friedrich-Alexander-University of Erlangen-Nuremberg (FAU), 91054 Erlangen, Germany; matthias.borutta@uk-erlangen.de (M.B.); gabriela.siedler@uk-erlangen.de (G.S.); bastian.volbers@uk-erlangen.de (B.V.); stefan.schwab@uk-erlangen.de (S.S.)

[*] Correspondence: iris.muehlen@uk-erlangen.de; Tel.: +49-(0)9131-85-44847

**Abstract:** Background: Innovative automated perfusion software solutions offer support in the management of acute stroke by providing information about the infarct core and penumbra. While the performance of different software solutions has mainly been investigated in patients with successful recanalization, the prognostic accuracy of the hypoperfusion maps in cases of futile recanalization has hardly been validated. Methods: In 39 patients with acute ischemic stroke (AIS) due to large vessel occlusion (LVO) in the anterior circulation and poor revascularization (thrombolysis in cerebral infarction (TICI) 0-2a) after mechanical thrombectomy (MT), hypoperfusion analysis was performed using three different automated perfusion software solutions (A: RAPID, B: Brainomix e-CTP, C: Syngo.via). The hypoperfusion volumes (HV) as Tmax > 6 s were compared with the final infarct volumes (FIV) on follow-up CT 36–48 h after futile recanalization. Bland–Altman analysis was applied to display the levels of agreement and to evaluate systematic differences. Based on the median hypoperfusion intensity ratio (HIR, volumetric ratio of tissue with a Tmax > 10 s and Tmax > 6 s) patients were dichotomized into high- and low-HIR groups. Subgroup analysis with favorable (<0.6) and unfavorable (≥0.6) HIR was performed with respect to the FIV. HIR was correlated to clinical baseline and outcome parameters using Pearson's correlation. Results: Overall, there was good correlation without significant differences between the HVs and the FIVs with package A (r = 0.78, *p* < 0.001) being slightly superior to B and C. However, levels of agreement were very wide for all software applications in Bland-Altman analysis. In cases of large infarcts exceeding 150 mL the performance of the automated software solutions generally decreased. Subgroup analysis revealed the FIV to be generally underestimated in patients with HIR ≥ 0.6 (*p* < 0.05). In the subgroup with favorable HIR, however, there was a trend towards an overestimation of the FIV. Nevertheless, packages A and B showed good correlation between the HVs and FIVs without significant differences (*p* > 0.2), while only package C significantly overestimated the FIV (−54.6 ± 56.0 mL, *p* = 0.001). The rate of modified Rankin Scale (mRS) 0–3 after 3 months was significantly higher in favorable vs. unfavorable HIR (42.1% vs. 13.3%, *p* = 0.02). Lower HIR was associated with higher Alberta Stroke Program Early CT Score (ASPECTS) at presentation and on follow-up imaging, lower risk of malignant edema, and better outcome (*p* < 0.05). Conclusion: Overall, the performance of the automated perfusion software solutions to predict the FIV after futile recanalization is good, with decreasing accuracy in large infarcts exceeding 150 mL. However, depending on the HIR, FIV can be significantly over- and underestimated, with Syngo showing the widest range. Our results indicate that the HIR can serve as valuable parameter for outcome predictions and facilitate the decision whether or not to perform MT in delicate cases.

**Keywords:** ischemic stroke; perfusion CT; automated CT perfusion software; artificial intelligence; mechanical thrombectomy

## 1. Introduction

As mechanical thrombectomy (MT) has gained central priority in the treatment of acute ischemic stroke due to large vessel occlusion (LVO), awareness of salvageable brain tissue plays a key role [1–3]. The fate of the ischemic tissue depends on the severity and duration of hypoperfusion [4]. The penumbra concept describes brain tissue at risk of infarction, which may remain viable for several hours after an ischemic event due to the collateral arteries that supply the penumbral zone. Knowledge about the penumbra is especially important to identify patients who can benefit from reperfusion therapies in the acute phase or even in an extended time window. However, penumbra analysis is quite challenging, and many approaches have been proposed so far [5–7].

Recently, several automated perfusion applications have been developed to offer support in this time-sensitive diagnostic process. The aim is to improve the diagnostic and prognostic accuracy, provide a standardized workflow, and ensure rapid clinical decision making and patient triage. This is especially valuable to practitioners with limited expertise in the diagnosis of stroke. Automated perfusion software applications could support communication and collaboration between referring sites and endovascular therapy (EVT) centers, facilitate appropriate patient transfer to EVT-treating centers, and eliminate unnecessary transfers or delays [8–12].

To determine the volume of the hypoperfused tissue (HV), the time when the residue function reaches its maximum (Tmax) has become the preferred parameter [13,14]. It is routinely obtained from CTP when deconvolution-based postprocessing is used. However, the agreement of the hypoperfusion maps of different automated software solutions with respect to the follow-up infarct after futile recanalization has hardly been investigated. Furthermore, Tmax > 6 s lesions include regions with variable degrees of hypoperfusion. Severely hypoperfused tissue with a large fraction of delay > 10 s has been associated with poor outcome [15,16]. Tissue with a higher percentage of less prolonged blood flow, however, may be at lower risk of infarction. The hypoperfusion intensity ratio (HIR), which represents the proportion of Tmax > 6 s lesion with a Tmax > 10 s delay, has been found to be an indicator of the collateral circulation [17]. It helps to identify tissue with more severely reduced cerebral blood flow, which has a potential impact on the evolution of the acute ischemic lesion [18]. Thus, this parameter could serve as an independent outcome predictor. The aim of our study was to compare the hypoperfusion results of three fully automated perfusion software applications with respect to the follow-up infarct volume (FIV) and to evaluate the importance of the HIR as a prognostic factor. Therefore, we chose a patient population with acute LVO in the anterior circulation in which endovascular thrombectomy with aspiration and/or stent retriever remained unsuccessful.

## 2. Materials and Methods

### 2.1. Study Population

We performed a retrospective cohort study of 60 consecutive patients with acute ischemic stroke (AIS) due to anterior circulation LVO and futile MT (TICI 0-2a). Data were obtained from our institutional stroke database at our comprehensive stroke center between January 2015 and December 2019. Study inclusion criteria contained AIS due to LVO (internal carotid artery and middle cerebral artery (M1, M2)), CT perfusion imaging at presentation, MT with a thrombolysis in cerebral infarction (TICI) 0-2a recanalization, and follow-up CT 24–36 h after MT.

Patients with missing perfusion data (n = 15), poor image quality due to severe motion artifacts (n = 1), unreliable maps (bi-hemispheric hypoperfusion, n = 1), missing follow-up imaging (n = 2), and severe complications during the intervention or as a consequence

of intravenous thrombolysis (n = 2) were excluded from the study. As an additional perfusion CT is not routinely performed in patients who are referred to our institution for thrombectomy from external hospitals, missing perfusion data was the main reason to exclude patients from the study. Thirty-nine patients were enrolled in the final analysis. The selection process is illustrated by a flowchart (Figure 1).

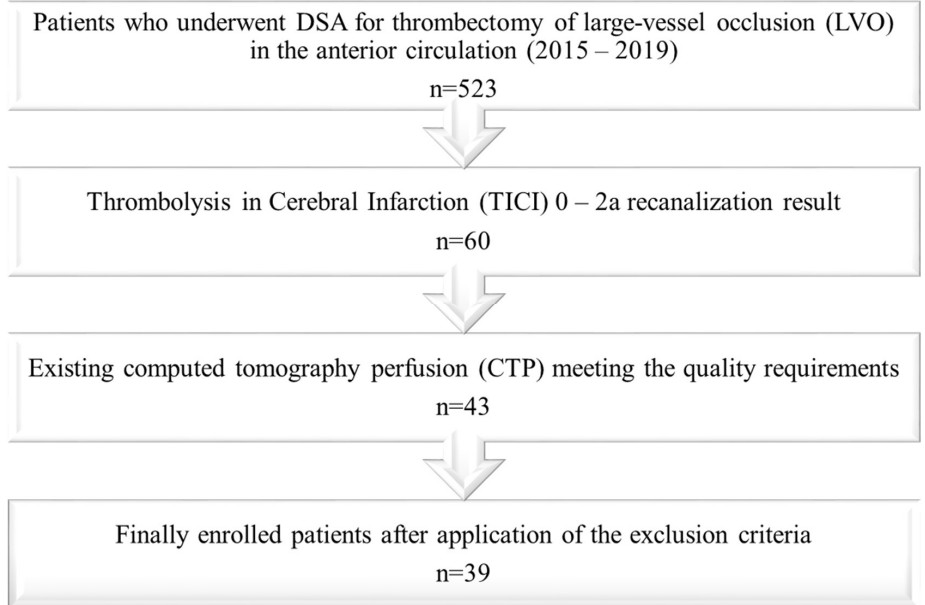

**Figure 1.** Flow chart illustrating the patient selection process.

The study was performed in compliance with the local ethics committee. All subjects or next of kin gave informed consent according to legal requirements.

### 2.1.1. Imaging and Image Reconstruction

CT was performed using a 128-section scanner (Somatom Definition AS+; Siemens Healthineers, Forchheim, Germany). A multimodal stroke protocol was used in the following order: thin-section non-enhanced computed tomography (NECT), computed tomography perfusion (CTP), computed tomography angiography (CTA).

NECT was performed in caudocranial direction (120 kV) and reconstructed with a slice thickness of 5 mm using a J30 kernel. Follow-up NECT was performed between 24 and 36 h after MT.

The imaging parameters for VPCT were as follows: 80 kV, 180 mAs, collimation 128 × 0.6 mm, rotation time 0.3 s. Pulsed full-rotation scan with 35 scans over 68 s and a scan delay of 4 s was used. 30 mL of iodinated contrast agent (Imeron 400; Bracco Imaging, Konstanz, Germany) followed by 50 mL of saline flush was administered intravenously at a rate of 5 mL/s using a double-piston power injector (Medtron, Saarbruecken, Germany). Brain coverage was 9.6 cm.

Arterial head and neck CTA was performed from the aortic arch to the cranial vertex (120 kV, 160 mAs, collimation 128 × 0.6 mm, rotation time 0.3 s). 60 mL of iodinated contrast agent (Imeron 400; Bracco Imaging, Konstanz, Germany) followed by 50 mL of saline flush was injected intravenously at a rate of 5 mL/s. Monitoring started with a delay of 5 s. Bolus tracking was performed in the descending thoracic aorta with a fixed start delay of 4 s after exceeding 100 HU.

### 2.1.2. Imaging Analysis

Baseline NECT (axial reconstructions, slice thickness 4.8 mm) was assessed for the presence of early ischemic changes defined by the Alberta Stroke Program Early CT Score (ASPECTS, range 0–10).

On CTA the site of occlusion was identified.

Follow-up CT scans 36–48 h after MT were assessed for the extent and location of infarction using a semiautomated segmentation method (syngo.via, Siemens, Erlangen, Germany). Midline shift was measured at the level of the septum pellucidum.

Imaging analysis was performed by two board-certified readers (6 and 8 years of experience in stroke imaging) in consensus, blinded to the initial CTP results and clinical outcome.

### 2.1.3. Automated CT Perfusion Software

The CTP data were postprocessed using three different software applications:

Package A: RAPID (iSchema View inc, Menlo Park, CA, USA).

Package B: Brainomix e-CTP (Brainomix Ltd., Oxford, UK).

Package C: Syngo.via CT Neuro Perfusion VB30 (Siemens Healthineers, Erlangen, Germany).

All software packages perform automated registration, segmentation and motion correction, and use a delay-insensitive algorithm. Automated calculation of ischemic core, hypoperfusion, and perfusion mismatch is provided.

Default settings for the determination of ischemic core differ among the software applications: RAPID and Brainomix define the ICV as the volume of tissues with at least 70% reduction in cerebral blood flow relative to the unaffected contralateral cerebral hemisphere (rCBF < 30%); Syngo works with a threshold of 80% (rCBF < 20%).

Hypoperfusion is defined as the time to the maximum of the residue function obtained by deconvolution exceeding 6s (Tmax > 6 s) for all software packages.

Core–perfusion mismatch is defined as the difference between the Tmax > 6 s lesion volume and the ischemic core volume. Mismatch ratio is calculated as the ratio between the Tmax > 6 s lesion volume and the core volume.

HIR represents the volumetric ratio of tissue with Tmax > 10 s and Tmax > 6 s.

### 2.2. Statistical Analysis

For statistical data analysis commercial software (SPSS 20, IBM, Chicago, IL, USA) was used. Results are described as mean ± standard deviation or as median with interquartile range. A P value <0.05 was considered statistically significant. The Pearson correlation coefficient was used to examine the relationship between FIV and HV for each software package. CI was set at 95%. The paired sample t test (for normally distributed data) was utilized to compare the volumetric differences between HV on automated perfusion mapping and FIV on follow-up CT for each setting. Variables were checked for normal distribution using the Kolmogorov–Smirnov and Shapiro–Wilk test. In addition, Bland–Altman plots were used to visualize the agreement of HV with FIV. For subgroup analysis, the study collective was dichotomized on the basis of the median HIR (volumetric ratio of tissue with a Tmax > 10 s and Tmax > 6 s). Subgroup analysis with high (≥0.6) and low (<0.6) HIR with respect to the FIV was performed using the paired sample t test. HIR was correlated to radiological and clinical baseline and outcome data using Pearson correlation.

### 3. Results

#### 3.1. Baseline Clinical and Outcome Results

Mean age was 76 (±11) years with 59% being female. Median pre-stroke mRS was 0 (0–3). Median NIHSS score at presentation was 17 (13–20). Mean duration of symptoms at the time of multimodal CT was 240 (±228) min. In 16 cases (41%) symptom onset was unclear. In these cases, the last-seen-normal time point was considered as onset. The occlusion was located in the left hemisphere in 20 cases (51.3%). The occlusion sites were at the internal carotid artery (ICA) (12.8%), Carotid-T (33.3%), M1-segment (35.9%), and M2-segment (18%). ICA stenosis was present in 5 cases (12.8%). Median ASPECTS on initial CT was 8 (7–10).

In 29 cases (74.4%), mechanical thrombectomy was accompanied by a bridging therapy with intravenous thrombolysis. In 33 patients (84.6%), the intervention was conducted under

general anesthesia. Mean procedure time was 146 (±108) min with a median of 4 (2–6) recanalization attempts. ICA stenting was performed in 3 cases (7.8%). A combination of aspiration and stent retriever devices was used in most of the cases (74.4%). Aspiration alone was only performed in 2 cases (5.1%). In 5 cases (12.8%), the thrombus could not be reached due to difficult vascular access conditions, and in another 12 patients (30.8%) it could not be passed due to its hard consistency. In 56.4%, the thrombus could be reached and passed, but several thrombectomy attempts remained unsuccessful either due to very hard thrombi, underlying stenosis, or chronic vessel occlusion. Final TICI score was 0 in 25 patients (64.1%), 1 in patients (10.3%), and 2a in 10 patients (25.6%). Median FIV was 180 (90–248) ml with a median ASPECTS of 3 (1–5). A midline shift exceeding 5 mm was present in 14 cases (35.9%). Median mRS after 3 months was 5 (4–6). Median reduction in the ASPECTS between initial CT and follow-up CT was 5 (3–7). Two patients had died prior to follow-up CT. Patient characteristics, clinical parameters, and outcome data are summarized in Table 1.

**Table 1.** Patient characteristics, clinical parameters, and outcome data.

| **Patient Demographics** | |
|---|---|
| Number of patients, n | 39 |
| Age, mean (SD), [years] | 76 (±11) |
| Female, n (%) | 23 (59) |
| Clinical characteristics | |
| Hypertonus, n (%) | 29 (74.3) |
| Diabetes, n (%) | 10 (25.6) |
| Hypercholesterolemia, n (%) | 16 (41) |
| Arterial fibrillation, n (%) | 19 (48.7) |
| Smoking, n (%) | 4 (10.3) |
| Admission National Institutes of Health Stroke Scale (NIHSS) score, median (IQR) | 17 (13–20) |
| Onset-to-imaging time, mean (SD), [min] | 240 (±228) |
| Prestroke modified Rankin Scale (mRS), median (IQR) | 0 (0-3) |
| Imaging characteristics | |
| Location of occlusion in left hemisphere, n (%) | 20 (51.3) |
| Site of occlusion, n (%) | |
| Internal Carotid Artery (ICA) | 5 (12.8) |
| Carotid T | 13 (33.3) |
| M1 segment | 14 (35.9) |
| M2 segment | 7 (18) |
| Clot Burden Score (CBS), median (IQR) | 5 (3–7) |
| ICA stenosis, n (%) | 5 (12.8) |
| Admission Alberta Stroke Program Early CT Score (ASPECTS), median (IQR) | 8 (7–10) |
| Treatment characteristics | |
| Bridging therapy with intravenous tPA, n (%) | 29 (74.4) |
| General anesthesia, n (%) | 33 (84.6) |
| procedure time, mean (SD), [min] | 146 (±108) |
| time interval from CTP to the end of the procedure, mean (SD), [min] | 194 (±113) |
| recanalization attempts, median (IQR) | 4 (2–6) |
| ICA stenting, n (%) | 3 (7.8) |
| recanalization technique, n (%) | |
| Stent retriever | 8 (20.5) |
| aspiration | 2 (5.1) |
| combination of both | 29 (74.4) |
| thrombus reached, n (%) | 34 (87.2) |
| thrombus passed, n (%) | 22 (56.4) |
| Outcome characteristics | |
| TICI, n (%) | |
| 0 | 25 (64.1) |
| 1 | 4 (10.3) |
| 2a | 10 (25.6) |
| Follow-up infarct volume, median (IQR), [mL] | 180 (90–248) |
| Follow-up ASPECTS, median (IQR) | 3 (1–5) |
| 3-month mRS, median (IQR) | 5 (4–6) |

*3.2. Perfusion Results*

Highest median hypoperfusion volumes were indicated by Syngo (195 mL, 142–239), lowest by Brainomix (158 mL, 103–215). No significant differences and good correlation of

FIV and HV were observed for all software applications ($p > 0.1$). Results are summarized in Table 2 and illustrated by Bland–Altman plots (Figure 2).

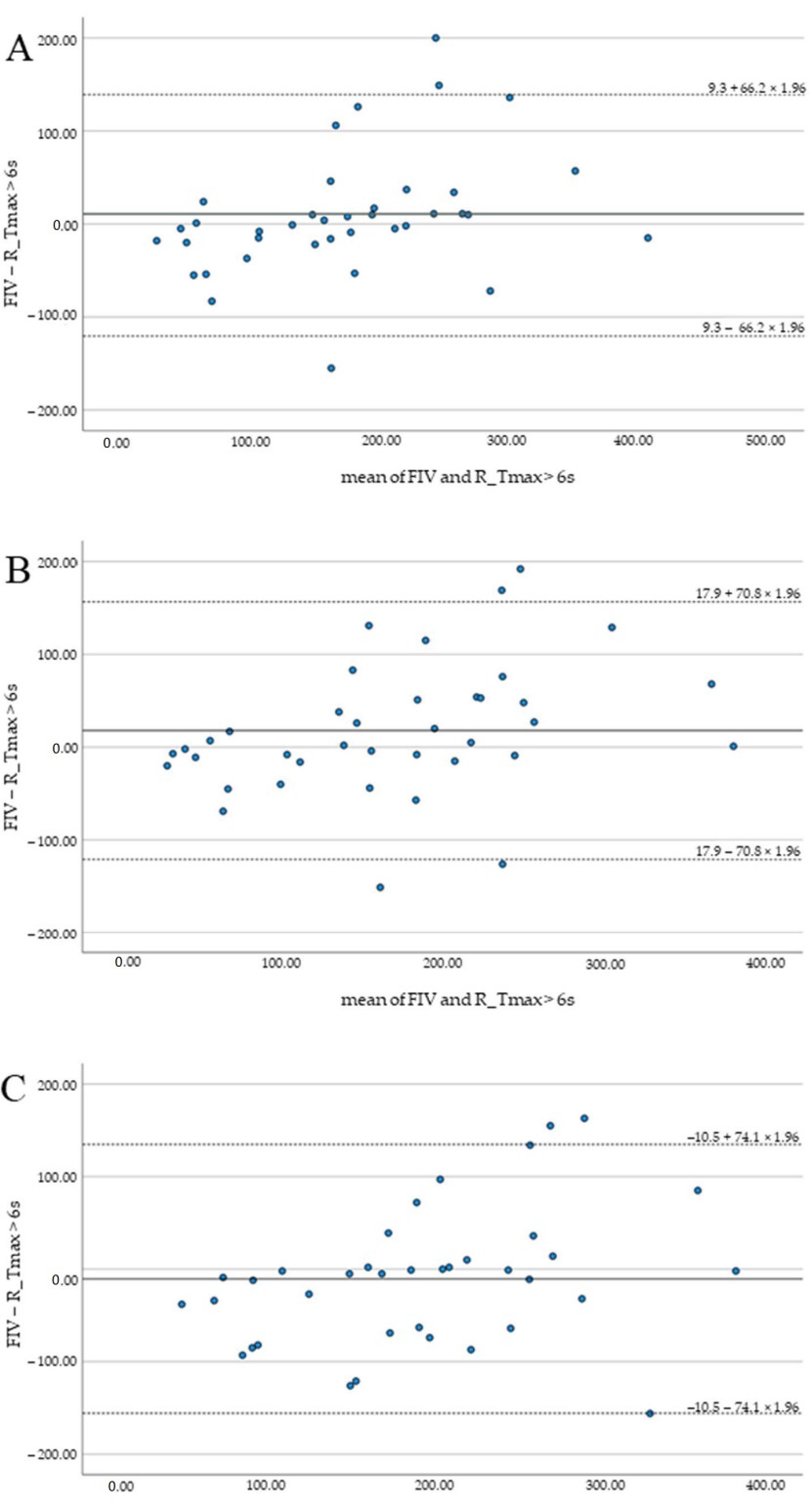

**Figure 2.** Bland–Altman plots illustrating the agreement of FIV and HV for RAPID (**A**), Brainomix (**B**), and Syngo (**C**).

**Table 2.** Performance characteristics of the RAPID, Brainomix, and Syngo software—agreement of the hypoperfusion volumes (HVs) and the follow-up infarct volumes.

|  | **RAPID** | **Brainomix** | **Syngo** |
|---|---|---|---|
| Mean difference (SD) of FIV and HV (mL) | 9.3 (66.2) | 17.9 (70.8) | −10.5 (74.1) |
| Pearson correlation (r) of FIV and HV | 0.78 | 0.74 | 0.70 |

The HVs differed significantly between RAPID and Syngo ($-21.7 \pm 37.9$ mL, $p < 0.001$) as well as Brainomix and Syngo ($-28.9 \pm 45.1$ mL, $p < 0.001$), while there were no significant differences between RAPID and Brainomix ($-8.3 \pm 41.0$ mL, $p = 0.22$). The same could be observed for the mismatch volumes, which did not show significant differences between RAPID and Brainomix ($3.8 \pm 43.5$ mL, $p = 0.29$), but between Brainomix and Syngo ($-27.7 \pm 46.3$ mL, $p < 0.001$) as well as RAPID and Syngo ($-25.3 \pm 41.4$ mL, $p < 0.001$). However, correlation of the mismatch volumes was high among all software packages ($r > 0.8$, $p < 0.001$). All software packages showed small differences (between 0.9 and 3.1 mL) in the indicated ischemic core volumes (ICV) without significant differences ($p > 0.3$). Median HIR was 0.6 (0.4–0.7) for all software applications. Figure 3 shows the automatically generated perfusion results by packages A, B, and C of a patient with M1-occlusion of the right hemisphere. The FIV on follow-up CT 24 h after futile recanalization is shown on Figure 4.

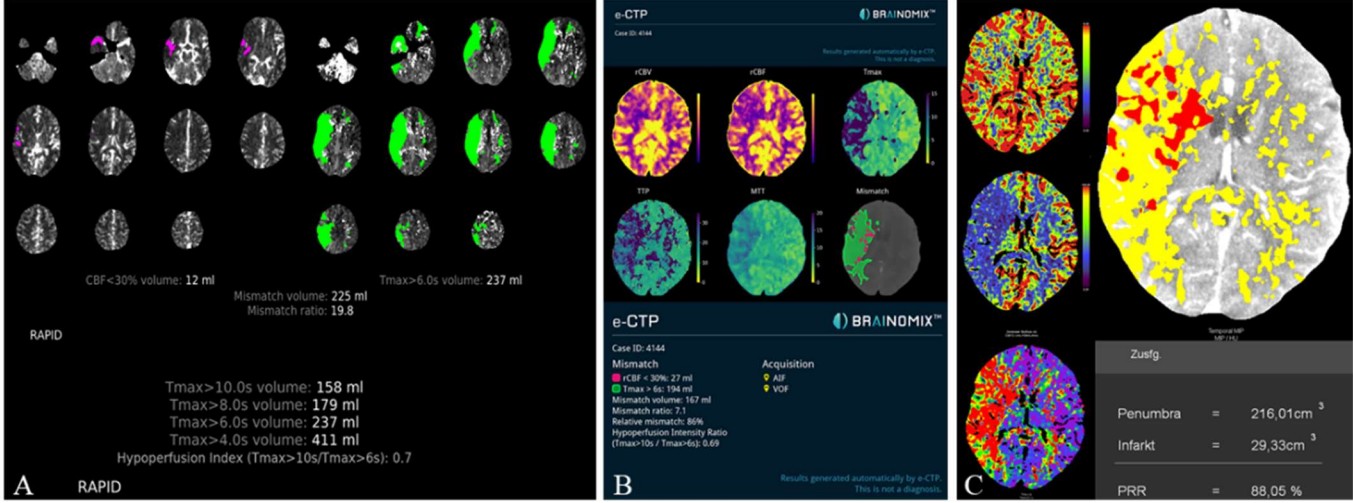

**Figure 3.** Illustrative Perfusion results of a patient with M1-occlusion of the right hemisphere (RAPID (**A**), Brainomix (**B**), and Syngo (**C**)). Packages A and B provide automated calculation of the HIR, which can be directly read from the perfusion maps. For Syngo, the HIR was calculated with a result of 0.7.

HIR values were below the median HIR value (0.6) in 19 patients (48.7%). In 20 patients (51.3%), HIR was equal to or above 0.6. Subgroup analysis revealed significant differences between FIV and HV for HIR $\geq 0.6$ ($p < 0.05$) for RAPID and Syngo. All software applications underestimated the FIV—Brainomix by 18.6 mL (the least) and RAPID by 35.1 mL (the most). For HIR < 0.6, good correlation and no significant differences were observed with RAPID ($-13.8 \pm 62.3$, $p = 0.26$; $r = 0.74$) and Brainomix ($17.2 \pm 59.3$, $p = 0.22$; $r = 0.83$), but Syngo still showed significant differences with the widest range between over- and underestimation (HIR $\geq 0.6$: $31.2 \pm 65$ mL; HIR <0.6: $-54.6 \pm 56$ mL; Figure 5). Brainomix was the only one to underestimate the FIV in the subgroup with HIR < 0.6.

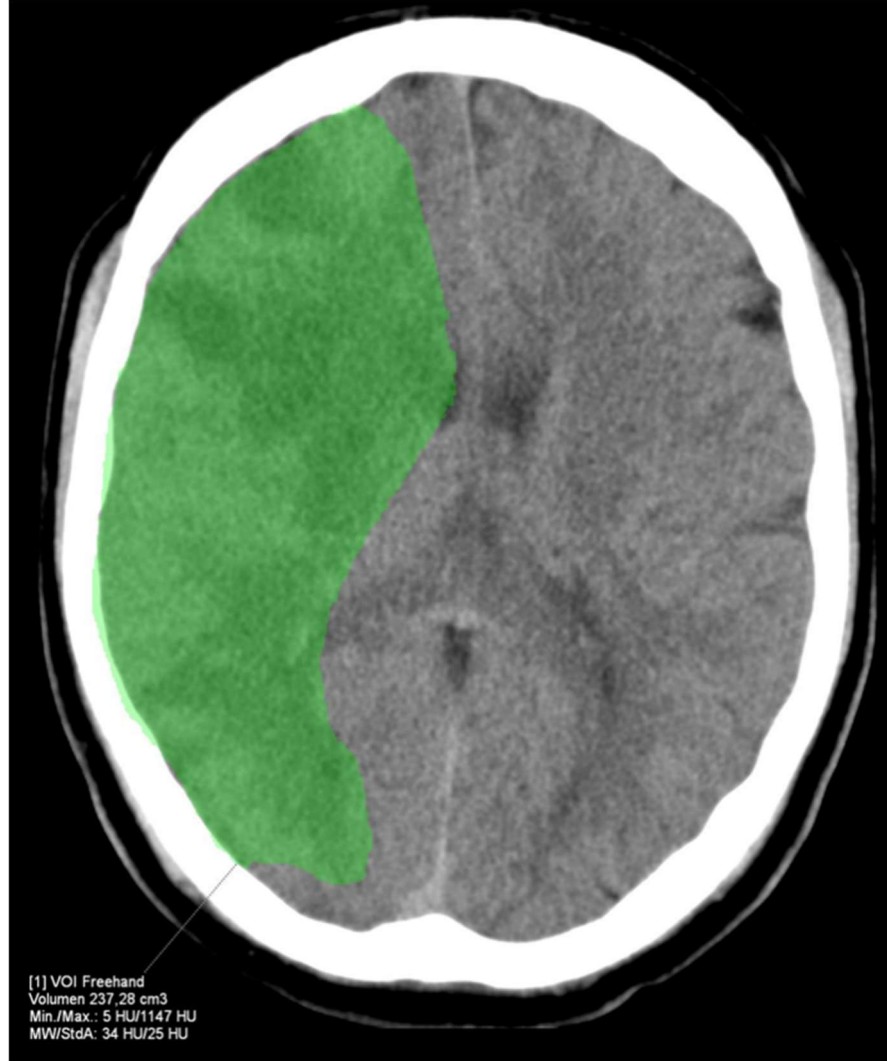

**Figure 4.** FIV of the example patient from Figure 3 24 h after futile recanalization. The FIV corresponds well with the HV indicated by RAPID.

Patients with higher HIR had lower ASPECTS on presentation (r = −0.6, *p* < 0.001) and on follow-up CT (r = −0.4, *p* < 0.05), higher FIV (r = 0.5, *p* < 0.05), and a higher risk of malignant edema with a midline shift >5 mm (r = 0.6, *p* < 0.001). There was a trend toward better clinical outcome in the subgroup of HIR < 0.6 (mRS at 3 months 4 (3–5) versus 5 (4–6)). However, statistical significance was barely missed. mRS 3 was more frequent in the HIR < 0.6 subgroup (42.1% versus 13.3%, *p* = 0.02). Due to our patient collective with a TICI 0-2a recanalization result, there was no patient with mRS ≤ 2 at 3 months.

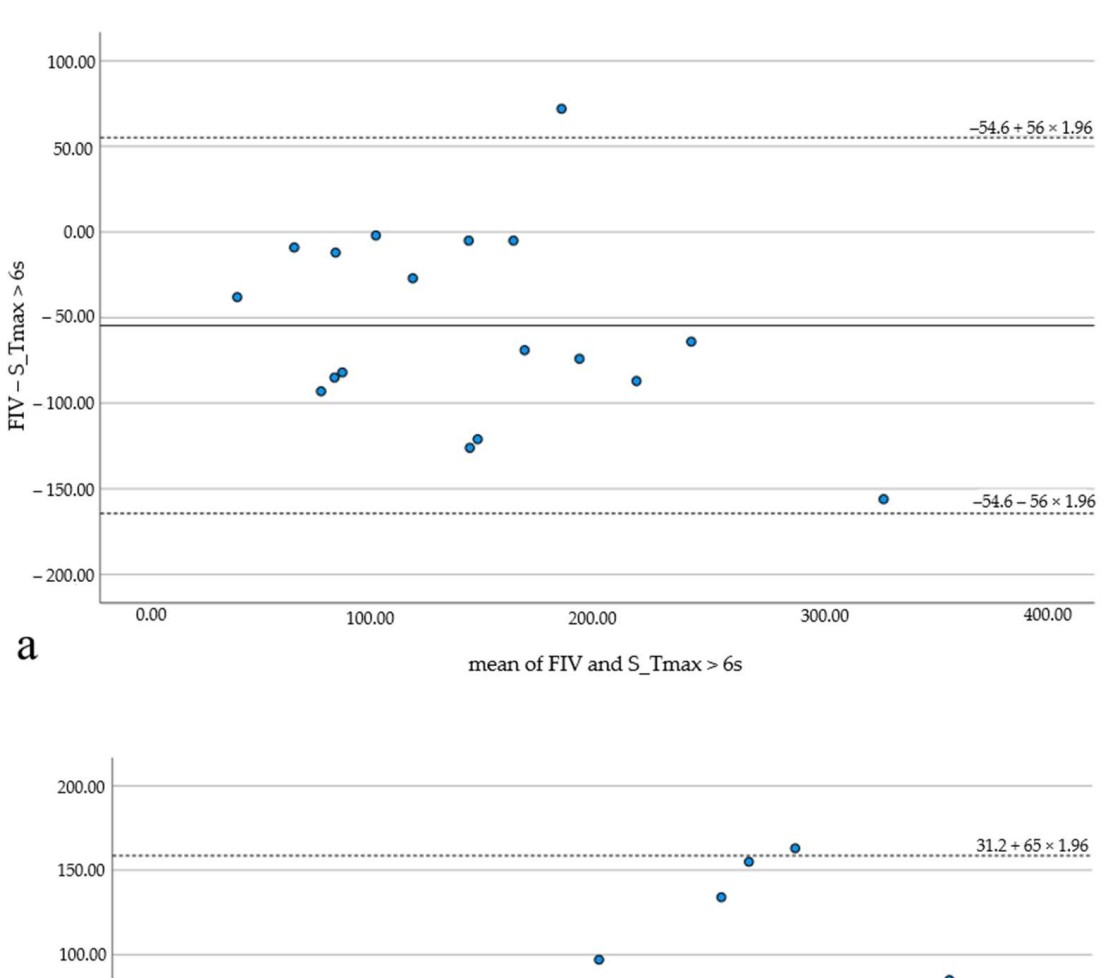

**Figure 5.** Bland–Altman plots illustrating the results of the subgroup analysis (HIR ≥ 0.6 and HIR < 0.6) with package C. The FIV is significantly overestimated in the subgroup with favorable HIR (**a**), while it is significantly underestimated in the subgroup with unfavorable HIR (**b**).

## 4. Discussion

The aim of our study was to investigate the prognostic accuracy of CTP summary maps in predicting the FIV in patients with acute ischemic stroke after futile MT and to evaluate the significance of the HIR, analyzing three different automated perfusion software applications. To our knowledge, this is the first study comparing the HIR of different software packages. An advantage of the study is that all examinations were performed on the same CT scanner with a standard CTP protocol and that the definition of hypoperfusion as Tmax > 6 s is the same for all software applications. Nevertheless,

HVs differed significantly between Syngo and the other two packages, while RAPID and Brainomix did not show significant differences among each other. These differences may be due to the vendor specific model used for the calculation of the perfusion maps. In addition, factors such as dispersion correction, additional noise correction, or "smoothing" may have influenced the results. Similar to the results of Austein et al., the highest median HVs were indicated by Syngo. Considering the median absolute differences between HV and FIV, RAPID and Brainomix underestimated the infarct volumes, while only Syngo overestimated the FIV. Brainomix underestimated the FIV more than RAPID. The smallest median absolute difference was indicated by RAPID. Although there was good correlation of the HVs and FIVs for all software applications without significant differences, Bland Altman analysis revealed that overall, the levels of agreement were very wide. Analogous to the study of Austein et al., who observed that deviation of the ischemic core volume from FIV increased with increasing core volumes, we noticed that deviation of the HV from FIV increased with increasing HVs [9]. In cases with very large infarcts (>150 mL), FIV was systematically underestimated on CTP outputs. This might be due to partial reperfusion of widely infarcted tissue via leptomeningeal collaterals, especially in an extended time window. Low ASPECTS on initial CT should be considered in these cases.

However, automated perfusion results might not only be confounded by large infarcts, but also by motion artifacts or poor contrast bolus arrival with the consequence of unreliable perfusion maps [19]. Thus, perfusion outputs should always be double-checked by an experienced neuroradiologist. In contrast to the HVs, median HIR did not differ between the software packages. In accordance with the studies of Rao et al., subgroup analysis showed that, depending on the HIR, infarct volume can be significantly over- and underestimated [20]. Thus, HIR has an impact on the evolution of the ischemic lesion. Our results correspond to previous studies, which could show that lower HIR is associated with a smaller infarct growth and better collateral circulation [17,18,21–23]. In the group of favorable HIR, FIV was rather overestimated by the perfusion software solutions with the exception of Brainomix, while it was generally underestimated in the group of unfavorable HIR. Package C showed the widest range between over- and underestimation. Both over- and underestimation of the FIV may have important consequences in the management of acute stroke, especially in cases where the decision whether to perform MT or not is not obvious. Such delicate cases particularly apply to patients with very advanced age or in whom secondary referral to a thrombectomy center involves delays for transportation [21,24–26]. HIR might be helpful in assessing the situation and making rational decisions in this time-sensitive phase. High HIR could limit the therapeutic potential of reperfusion therapies, while low HIR could be an indicator of better prognosis. This is in accordance with the studies of Keenan et al., who found that the volume of hypoperfused tissue associated with poor outcome differs in dependence on the Tmax threshold [16]. According to the studies of Murray et al., HIR is associated with the severity of stroke at baseline and with the risk to develop malignant edema after futile recanalization [27]. Analogously, we found that ASPECTS at baseline and on follow-up imaging and the risk of developing malignant edema were significantly associated with the HIR. The median HIR in our study lies within the range of the cutoff values applied in previous studies (0.4–0.7) [17,18,21–27]. However, further research is necessary to define a generally valid cutoff value for HIR.

*Our Study Has Several Limitations*

The first is related to the retrospective design of the study and the relatively small patient collective. However, considering the exclusive patient population, the number of included patients appears appropriate [19].

The second is related to the use of the follow-up CT at 24–36 h as a gold standard for the definition of the FIV, as the exact delineation of the infarct may be challenging on NECT. To moderate this limitation, analysis was performed by two experienced readers in consensus, blinded from the initial CTP results.

The third is related to the fact that the impact of a bridging therapy with intravenous tPA (74.4%) or of partial reperfusion after MT (25.6% with a TICI 2a recanalization result) has not been considered.

The last is related to the perfusion maps. Hypoperfusion might be overestimated by the software due to artifacts in the skull base mimicking hypoperfusion abnormalities. Results might also be confounded by limitations to the arterial flow, such as chronic carotid stenosis or low cardiac output. However, cases with unreliable maps, such as bi-hemispheric hypoperfusion or severe artifacts were excluded from the study a priori.

These potential sources of error show, however, that the automatically generated perfusion maps obligatorily need to be double-checked by a radiologist.

## 5. Conclusions

Good accuracy of the perfusion maps in predicting the FIV after futile MT was found for all software packages. RAPID was the one with highest precision. Subgroup analysis showed that, depending on the HIR, infarct volume can be significantly over- or underestimated, with Syngo showing the widest range. Our results indicate that the HIR can provide valuable additional information on the evolution of the ischemic lesion and functional outcome with the potential to facilitate decision making whether or not to perform MT in delicate cases.

**Author Contributions:** Conceptualization, I.M. and T.E.; Data curation, I.M., M.B., G.S., S.H. and M.K.; Formal analysis, I.M.; Investigation, I.M. and P.H.; Methodology, I.M. and P.H.; Project administration, T.E.; Resources, T.E.; Software, I.M. and P.H.; Supervision, S.S. and A.D.; Writing—original draft, I.M.; Writing—review & editing, B.V. and A.D. All authors have read and agreed to the published version of the manuscript.

**Funding:** This research received no external funding.

**Institutional Review Board Statement:** The study was conducted in accordance with the Declaration of Helsinki and performed in compliance with the local ethics committee (Friedrich-Alexander University of Erlangen-Nuremberg, Date of approval: 2 November 2020, protocol code: 33_20 B).

**Informed Consent Statement:** All subjects or next of kin gave informed consent for inclusion in the study and publication of this paper according to legal requirements.

**Data Availability Statement:** The data presented in this study are available on request from the corresponding author.

**Conflicts of Interest:** The authors declare no conflict of interest. The Department of Neuroradiology, University of Erlangen-Nuremberg, has a research agreement with Siemens Healthineers (Forchheim, Germany), with iSchemaView, Inc. (Menlo Park, CA, USA) and with Brainomix Ltd. (Oxford, UK).

## Abbreviations and Acronyms

| | |
|---|---|
| AIS | acute ischemic stroke |
| LVO | large vessel occlusion |
| TICI | Thrombolysis in cerebral infarction |
| MT | mechanical thrombectomy |
| HV | hypoperfusion volume |
| FIV | follow-up infarct volume |
| HIR | hypoperfusion intensity ratio |
| mRS | modified Rankin Scale |
| ASPECTS | Alberta Stroke Program Early CT Score |
| EVT | endovascular therapy |

| Tmax | time when the residue function reaches its maximum |
| --- | --- |
| CTP | computed tomography perfusion |
| NECT | non-enhanced computed tomography |
| VPCT | volume perfusion CT |
| CTA | computed tomography angiography |
| ICV | ischemic core volume |
| NIHSS | National Institutes of Health Stroke Scale |
| ICA | internal carotid artery |

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
