# Peer review of "Prognostic Accuracy of CTP Summary Maps in Patients with Large Vessel Occlusive Stroke and Poor Revascularization after Mechanical Thrombectomy—Comparison of Three Automated Perfusion Software Applications"

_tomography, doi:10.3390/tomography8030109_

Round 1
Reviewer 1 Report
The paper submitted by Muehlen et al. gives helpful information about The Prognostic accuracy of CTP summary maps in patients with large-vessel occlusive stroke and poor revascularization after mechanical thrombectomy. However, some sections are confused and should be carefully revised. In addition, the study population was done with a very short number of cases.Specific comments Since there are too many abbreviatures I recommend using a list of them at the beginning of the paper. In the abstract, some long sentences should be shortened The introduction section is too short, and as consequence, some concepts are missed. Line 51, please add some explanation since penumbra (ischemic penumbra is a relatively new concept). Moreover, the authors use a recent reference (7), where the importance of that concept is presented, then, redone. Line 52, lately does not include references from 2010, Line 55, please pay attention to long sentences. Line 56, please provide a short explanation of the "residue function" Line 62-64, it is obvious, please redone. Line 72, In this section, I miss how you performed the endovascular thrombectomy and some information about the study population Line 84-87, this sentence is quite confusing, please redo it. Line 90, please provide the reference of the ethical protocol and the informed consent (provide documents to the editor) Line 94, please revised the thickness, as well as the complete paragraph. Line 106, please check "cranial vertex", I am not sure about the correct terminology, is it included in the International Anatomical Nomenclature? Line 116, please describe where the site of occlusion was identified. Line 132, this sentence is not clear (the blood flow that can read the packages?). Lines 165-170, this paragraph describes some information that should be moved to the discussion section. In general, there is a lot of information that could be better explained in a table, not only in a figure. The authors must remind that articles are not only addressed to the specialist of the field, but also to general practitioners. Line 259, this point is quite important, and by studying this, the paper would be more solid. Line 289, it is quite important since the authors recognized the several limitations identified in this study Finally, I would like to know how frequently the authors apply this procedure so important in this life-threatening condition.
Author Response
Dear Reviewer,
Thank you very much for your review and very constructive remarks, which we
all addressed within the manuscript.
- Since there are too many abbreviations, I recommend using a list of them at the beginning of the paper.
Thank you for your comment. We included a list of abbreviations at the beginning of the paper.
- In the abstract, some long sentences should be shortened.
To make the abstract more readable we shortened some long sentences.
- The introduction section is too short, and as consequence, some concepts are missed. Line 51, please add some explanation since penumbra (ischemic penumbra is a relatively new concept). Moreover, the authors use a recent reference (7), where the importance of that concept is presented, then, redone. Line 52, lately does not include references from 2010, Line 55, please pay attention to long sentences. Line 56, please provide a short explanation of the "residue function" Line 62-64, it is obvious, please redone. Line 72, In this section, I miss how you performed the endovascular thrombectomy and some information about the study population.
Thank you for this remark. We revised the introduction by paying attention to your comments. We explicated the penumbra concept and the residue function in more detail and included a reference to pertinent earlier work of our group (Muehlen et al. Stroke. 2021; doi: 10.1161/STROKEAHA.121.035626).
- Line 84-87, this sentence is quite confusing, please redo it.
Thank you for your advice. We restructured this sentence.
- Line 90, please provide the reference of the ethical protocol and the informed consent (provide documents to the editor)
The approval of the study by the ethics committee and the blank version of patients’ informed consent are provided.
- Line 94, please revise the thickness, as well as the complete paragraph.
We admit that this paragraph was packed with many technical details and revised it to make it more readable and clear.
- Line 106, please check "cranial vertex", I am not sure about the correct terminology, is it included in the International Anatomical Nomenclature?
“vertex” is an officially used term included in the International Anatomical Nomenclature
Vertex (noun)
ver·â€‹tex | \ ˈvÉ™r-ËŒteks \
plural vertices\ ˈvər-​tə-​ˌsēz \ also vertexes
Definition of vertex
- (anat.): the top of the head
- Line 116, please describe where the site of occlusion was identified.
A description of the site of the occlusion was included in the “baseline results” section.
- Line 132, this sentence is not clear (the blood flow that can read the packages?)
The sentence was revised to make it better understandable.
- Lines 165-170, this paragraph describes some information that should be moved to the discussion section. In general, there is a lot of information that could be better explained in a table, not only in a figure.
Thank you very much for this comment.
We revised the paragraph, moved some information to the discussion section and included a table (table 2) to make results more clear.
- Line 259, this point is quite important, and by studying this, the paper would be more solid.
Thank you for that comment. This is an important issue, which we included in the discussion section.
- Finally, I would like to know how frequently the authors apply this procedure so important in this life-threatening condition.
The analysis of the generated perfusion results from automated perfusion software solutions is part of our routine clinical practice. At our institution we perform 150 to 200 endovascular thrombectomy procedures per year.
We very much hope that our study fulfils the standards of Tomography and that our revised manuscript meets the requirements for publication.
Yours sincerely,
Iris Mühlen
Reviewer 2 Report
Muehlen et. al. investigated the prognostic accuracy of the hypoperfusion maps of three different software solutions in patients with large-vessel occlusive stroke and futile recanalization after mechanical thrombectomy. Impressively, the RAPID software was the one with the highest precision, and HIR can provide valuable additional information on the evolution of the ischemic lesion with the potential to facilitate decision making whether or not to perform MT in delicate cases. The results are mostly supported by experimental data. However, the following revisions should be made prior to the acceptance.
- Page1, line 23, in the abstract, the authors claimed that “Subgroup analysis with favorable (< 0.6) and unfavorable (≥ 0.6) hypoperfusion intensity ratio”, the reason for cut-off setting and the details of subgroup analysis should be declared in the manuscript.
- Page 5, line 204, TICI score was 1 in ( ) patients. The number was lost.
- The time intervals between CTP and reperfusion should be added in Table 1.
- Page 10, line 281, the sentence “The more severe hypoperfusion is, the smaller might be the volume of the hypoperfused tissue to predict poor outcome” makes readers confused.
- Were all data sets checked for normality before analysis with the appropriate parametric or nonparametric tests? For non-normality data, median values and interquartile ranges should be calculated and presented rather than means and SDs.
- Page 4, line 170 -175. Which statistical methods were applied for these comparisons?
- The results and discussion parts should be better organized for easy understanding.

Author Response
Dear Reviewer,
Thank you very much for your review and very constructive remarks, which we addressed within the manuscript.
- Page1, line 23, in the abstract, the authors claimed that “Subgroup analysis with favorable (< 0.6) and unfavorable (≥ 0.6) hypoperfusion intensity ratio”, the reason for cut-off setting and the details of subgroup analysis should be declared in the manuscript.
Thank you very much for this remark. Analogously to previous studies (eg Olivot et al Stroke. 2014; doi:1161/STROKEAHA.113.003857), the cut-off was defined by the median of the hypoperfusion intensity ratio, which was 0.6 for all software applications. The median HIR in our study lies within the range of the cut-off values applied in previous studies (0.4 to 0.7). However, we realize that further research is necessary to define a generally valid cut-off for the HIR and we included this issue into the discussion. Additionally, subgroup analysis was explained in more detail.
- Page 5, line 204, TICI score was 1 in ( ) patients. The number was lost.
We apologize for this mistake. The number was supplemented.
- The time intervals between CTP and reperfusion should be added in Table 1.
As our study population consisted exclusively of patients with futile recanalization we cannot provide this time interval. Instead, we added the time interval between CTP and the end of the procedure.
- Page 10, line 281, the sentence “The more severe hypoperfusion is, the smaller might be the volume of the hypoperfused tissue to predict poor outcome” makes readers confused.
We understand your point of criticism and omitted this sentence.
- Were all data sets checked for normality before analysis with the appropriate parametric or nonparametric tests?
Prior to the application of a parametric or nonparametric test, normality was checked using the Kolmogorov-Smirnov and Shapiro-Wilk test. An example for the FIV is attached.
- Page 4, line 170 -175. Which statistical methods were applied for these comparisons?
For these comparisons the paired sample t-test (normally distributed data) and the Pearson correlation was applied.
- The results and discussion parts should be better organized for easy understanding.
Thank you for your advice. We revised the results and discussion section by clarifying the text and making it more readable.
We very much hope that our revised manuscript meets the requirements for publication in “Tomography”.
Yours sincerely,
Iris Mühlen

Reviewer 3 Report
I ABBREVIATIONS
According to Tomography | Instructions for Authors (mdpi.com)
Should be defined the first time they appear in each of three sections: the abstract; the main text; the first figure or table. When defined for the first time, the acronym/abbreviation/initialism should be added in parentheses after the written-out form.
Example : Please check and define abbreviations inserted in Table 1 and Figure 1
II REFERENCES
According to Tomography | Instructions for Authors (mdpi.com)
According to Journal Articles:
1. Author 1, A.B.; Author 2, C.D. Title of the article. Abbreviated Journal Name Year, Volume, page range.
Please check:
In some references, page ranges are incorrect. For example in ref 2 page rang 1219-1223 and in Ref 9 2311-7
lease check
Reference number 2
Abbreviated Journal Name .
Cohen JE, Rabinstein AA, Ramirez-de-Noriega F, Gomori JM, Itshayek E, Eichel R, Leker RR. Excellent rates of recanalization 344 and good functional outcome after stent-based thrombectomy for acute middle cerebral artery occlusion. Is it time for a para- 345 digm shift? Journal of Clinical Neuroscience. 2013; 20:1219-1223. doi: 10.1016/j.jocn.2012.11.011
Reference number 9
Year
Austein F, Riedel C, Kerby T, Meyne J, Binder A, Lindner T, Huhndorf M, Wodarg F, Jansen O. Comparison of Perfusion CT Software to Predict the Final Infarct Volume After Thrombectomy. Stroke. 2016 Sep; 47:2311-7. doi: 10.1161/STROKEAHA.116.013147.
Reference number 17
Volume
Wang CM, Chang YM, Sung PS, Chen CH. Hypoperfusion Index Ratio as a Surrogate of Collateral Scoring on CT Angiogram in Large Vessel Stroke. J Clin Med. 2021; 21;10:1296. doi: 10.3390/jcm10061296.
Reference number 27
Year, Volume, page range.
Monteiro A, Cortez GM, Greco E, Aghaebrahim A, Sauvageau E, Hanel RA. Hypoperfusion intensity ratio for refinement of elderly patient selection for endovascular thrombectomy. J Neurointerv Surg. 2021: neurintsurg-2020-017218. doi: 10.1136/neu-rintsurg-2020-017218.
Author Response
Dear Reviewer,
Thank you very much for your review and constructive remarks.
I ABBREVIATIONS
Abbreviations have been checked and defined. According to the suggestion of another reviewer, a list of abbreviations has been included at the beginning of the paper.
II REFERENCES
References have been checked and revised according to Tomography | Instructions for Authors
We very much hope that our study fulfils the standards of Tomography and that our revised manuscript meets the requirements for publication.
Yours sincerely,
Iris Mühlen
Round 2
Reviewer 1 Report
Please include some paragraphs explaining how this procedure can help general practitioners since in many instances they are the ones that see these patients and send them to the referral center
Author Response
Dear Reviewer,
Thank you very much for your constructive remarks, which we addressed within the manuscript.
Please include some paragraphs explaining how this procedure can help general practitioners since in many instances they are the ones that see these patients and send them to the referral center
Several Computer-assisted diagnosis (CAD) systems have already been implemented in the clinical routine in many domains of radiology. The purpose of CAD is to improve the diagnostic accuracy and the consistency of the radiologists’ image interpretation. Automated Imaging Software Packages for Stroke allow for quick visualization and analysis of brain physiology and fast detection of early ischemic changes. By providing clear and easy to interpret results on the site of vessel occlusion, early ischemic changes and salvageable brain tissue, they support timely, standardized interpretation of the multimodal stroke imaging. Thus, they can assist with rapid clinical decision making and patient triage. This is especially valuable to practitioners with limited expertise in the diagnosis of stroke. Furthermore, they can support communication and collaboration between referring sites and EVT centres, facilitate appropriate patient transfer to EVT treating centres and eliminate unnecessary transfers or delays.